# The Molecular Characterization of *bla*_NDM-1_-Positive *Acinetobacter baumannii* Isolated in Central Greece

**DOI:** 10.3390/microorganisms11102588

**Published:** 2023-10-19

**Authors:** Katerina Tsilipounidaki, Christos-George Gkountinoudis, Zoi Florou, George C. Fthenakis, Vivi Miriagou, Efthymia Petinaki

**Affiliations:** 1Faculty of Medicine, University of Thessaly, 41500 Larissa, Greece; tsilipoukat@gmail.com (K.T.); gountinoudis@gmail.com (C.-G.G.); zoi_fl@yahoo.gr (Z.F.); 2Veterinary Faculty, University of Thessaly, 43100 Karditsa, Greece; gcf@vet.uth.gr; 3Laboratory of Bacteriology, Hellenic Pasteur Institute, 11521 Athens, Greece; miriagou@pasteur.gr

**Keywords:** *Acinetobacter baumannii*, antibiotic resistance, *bla*
_NDM-1_, carbapenem, Greece

## Abstract

The objective of the present study is to report the detection and the molecular characterization of nine *bla*_NDM-1_-positive *Acinetobacter baumannii* isolates, which were isolated from patients in a tertiary care hospital in Central Greece from December 2022 to August 2023. The isolates were characterized by whole genome sequencing to obtain Pasteur multilocus sequencing typing (MLST) and to identify the *bla*_NDM-1_-environment, resistome, and virulence genes content. In silico MLST analysis showed that the isolates belonged to four different clones (STs 160, 2, 85, and 2493). All strains, apart from the *bla*_NDM-1_-gene, possessed at least eight different genes, encoding resistance to various antimicrobial agents. Whole genome sequencing revealed two different structures of the *bla*_NDM-1_ environment. The first, detected in ST160 strain, was identical with the Tn*125*, whereas the second, found in STs 2, 85, and 2493 was associated with Tn*7382.* To our knowledge, after a sole strain reported in 2016 and imported by a patient hospitalized in a Libyan hospital, this is the first report of the emergence of polyclonal *bla*_NDM-1_-positive *Acinetobacter baumannii* in Greece. Our findings re-emphasize the need to apply diligent surveillance protocols in order to limit the horizontal transfer of the *bla*_NDM-1_ gene to other *A. baumannii* clones or to other recipient strains.

## 1. Introduction

*Acinetobacter baumannii* (AB), a nonfermenting Gram-negative coccobacillus, is responsible for severe infections (septicaemia and pneumonia) in hospitalised patients [1]. People at high risk for *A. baumannii* infections are those with prolonged stays in Intensive Care Units (ICUs), who are under the support of ventilator or medical devices (e.g., catheters) [2,3]. In recent years, carbapenem-resistant *A. baumannii* (CRAB) isolates have received considerable attention due to their dissemination worldwide, combined with the limited therapeutic options in such cases. Specifically, apart from resistance to *β*-lactams, these microorganisms have also been found to be frequently resistant to aminoglycosides, tetracyclines, fluoroquinolones, colistin, and trimethoprim–sulfamethoxazole [4].

In carbapenem-resistant *A. baumannii*, the mechanisms of resistance to *β*-lactams can be typically classified into the following four groups: alterations in penicillin-binding proteins, loss of outer-membrane porins, overexpression of efflux pumps, and production of carbapenem-hydrolysing *β*-lactamases (carbapenemases) [5]. Until now, carbapenemases are the most prevalent mechanism occurring among such isolates and belong to Ambler class A (KPC), B (metallo-*β*-lactamases: NDM, VIM, GIM, SIM, and IMP), and D (oxacillinases: OXA-23, OXA-24/40, OXA-58, OXA-143, and OXA-235) [5,6,7,8].

According to WHONET Greece data, the rate of carbapenem-resistant *A. baumannii* ranges from 96.0 to 99.2% of all isolates, whereas *bla*_OXA-23_ remains the unique and predominant gene among such isolates [9]. The sole isolate carrier of both *bla*_OXA-23_ and *bla*_NDM-1_ was reported in 2016 to have been isolated from a patient who had been hospitalized for two months in a Libyan hospital [10]. On December 2022, in the University Hospital of Larissa (UHL) we recovered the first *bla*_NDM-1_-positive carbapenem-resistant *A. baumannii* isolate; subsequently, another eight such isolates were also recovered from hospitalised patients within a period of nine months. Although several studies have described the circulation of *bla*_NDM-1_-positive *A. baumannii* in various countries of the world, no *bla*_NDM-1_-positive carbapenem-resistant *A. baumannii* isolates have been identified in Greece since the first report [11,12,13,14,15,16,17,18,19,20,21,22,23,24,25,26].

The objective of the present study is to report the detection and the molecular characterization of nine *bla*_NDM-1_-positive *A. baumannii*, which were isolated from patients in a tertiary care hospital in Central Greece between December 2022 and August 2023.

## 2. Materials and Methods

### 2.1. Isolation of bla_NDM-1_ Acinetobacter baumannii

Nine *bla*_NDM-1_-positive *A. baumannii* isolates were studied in this work. All the isolates were recovered from clinical samples collected from patients admitted to the University Hospital of Larissa, a tertiary-care hospital in Central Greece. The isolates were recovered between December 2022 and August 2023 and correspond to 5.2% of the total *A. baumannii* isolates.

The identification and the susceptibility testing of the isolates were performed by the automated Vitek-2 system (BioMerieux, Marcy l’ Etoile, France). The Minimal Inhibitory Concentration (MICs) of imipenem and meropenem were confirmed by MIC test strips (Liofilchem, Roseto degli Abruzzi, Italy); the MIC to colistin was determined by the broth microdilution method (ComASP™ Colistin, Liofilchem, Roseto degli Abruzzi, Italy), according to the EUCAST guidelines (www.eucast.org) [27]. All isolates found to be phenotypically resistant to imipenem (MIC > 4 mg/L) and meropenem (MIC > 8 mg/L) were tested for the presence of carbapenemase-encoding genes: *bla*_OXA-23_, *bla*_OXA-58_, *bla*_KPC_, *bla*_VIM_, and *bla*_NDM_ by employing a relevant PCR followed by sequencing analysis [28,29].

### 2.2. Whole Genome Sequencing of bla_NDM-1_-Positive Acinetobacter baumannii

All the isolates were further characterized by Whole Genome Sequencing (WGS).

Originally, libraries were created using Ion Torrent technology and Ion Chef Work flows (Thermo Fisher Scientific, Waltham, MA, USA). The genomic DNA libraries underwent sequencing on the S5XLS system, and primary data analysis was carried out using Ion Torrent Suite (version 5.10.0). To assess the quality of the reads, FastQC software (version 0.11.9) was employed. Subsequently, the reads for each sample were assembled using the SPAdes genome assembler (version 3.15.5) with the default settings. The quality of the assembled genomes was evaluated using the Quast tool (version 5.2.0). The average coverage for each genome was determined using the mapPacBio tool from BBTools (available at https://sourceforge.net/projects/bbmap/; accessed on 27 August 2023). Gaps were addressed by sequencing the overlapping PCR-generated fragments.

The typing of isolates was performed using the online tool MLST 2.0 based on the Pasteur scheme. To identify the acquired antibiotic resistance genes within the assembled genomes, the online tool ResFinder-4.1 was utilized, with an identification threshold of 90% and a minimum gene length of 60%. Virulence factor characterization was conducted using the Virulence Factor Data Base. To investigate the genetic contexts of the *bla*_NDM-1_-encoding genes, a BLAST analysis was carried out, with only the results showing a high identity score (100% identity and ≥90% coverage) being considered. Furthermore, the genomes of our strains were compared with the available strains using Blast analysis.

### 2.3. Nucleotide Accession Numbers

The genomes of the *bla*_NDM-1_-positive *A. baumannii* isolates (A3232, A3496, A3557, A3569, A3570, A3593, A3597, A3629, and A3637) have been deposited in GenBank under BioProject accession PRJNA1018864.

## 3. Results

### 3.1. Antimicrobial Susceptibility Profiles

All *bla*_NDM-1_-positive *A. baumannii* displayed carbapenem resistance, with the MIC values to imipenem and meropenem exceeding 16 mg/L. In addition to carbapenem resistance, the nine strains also concurrently exhibited resistance to other antibiotics, specifically to amikacin (≥16 mg/L), ciprofloxacin (≥4 mg/L) and trimethoprim–sulphamethoxazole (≥320 mg/L). Eight strains were also found to be resistant to levofloxacin (≥4 mg/L), six of the strains were also found to be resistant to gentamicin (≥8 mg/L), six strains were resistant to colistin (≥16 mg/L), and three strains exhibited resistance to tobramycin (≥16 mg/L).

### 3.2. Multilocus Sequence Typing (MLST)

Based on the Pasteur MLST scheme, in silico analysis identified four Sequence Types (STs). One strain was classified to ST160 and three to each of the following STs, ST2 and ST85. The remaining two strains were characterized by a previously unidentified combination of alleles, specifically, *cpn60*:2, *fusA*:2, *gltA*:2, *pyrG*:1, *recA*:5, *rplB*:2, and *rpoB*:1, and were subsequently classified to a novel type, assigned as ST2493.

### 3.3. Identification of Resistance and Virulence Genes

WGS analysis revealed that the nine *bla*_NDM-1_ *A. baumannii* strains contained various genes responsible for resistance to various antimicrobial agents. These included seven genes associated with *β*-lactam resistance (*bla*_NDM-1_, *bla*_OXA-23_, *bla*_OXA-66_, *bla*_OXA-69_, *bla*_OXA-94_, *bla*_TEM-10_, and *bla*_ADC-25_), and among these, *bla*_NDM-1_, *bla*_OXA-23_, and *bla*_ADC-25_ were consistently identified among all nine strains. The analysis also revealed the presence of seven genes linked to aminoglycoside resistance (*aph(3′)-Ia*, *aph(6)-Id*, *aph(3′)-VI*, *aph(3″)-Ib*, *aac(3)-Ia*, *aadA1*, and *armA*), two genes conferring resistance to sulphonamides (*sul1* and *sul2*), macrolides (*mphE* and *msrE*), or tetracyclines (*tet(B)* and *tet39*), and one gene associated with the presence of resistance to trimethoprim (*dfrA1*) and quaternary ammonium resistance (*qacE*) (Figure 1).

In addition, WGS analysis revealed the presence of various genes encoding virulence factors. These included genes for exotoxin production, immune system modulation, and biofilm formation (*plc1*, *plc2*, *plcD*, *lpxA*, *lpxB*, *lpxC*, *lpxD*, *lpxL*, *lpxM*, *ompA*, *bap*, *adeG*, etc.).

### 3.4. Genetic Environment of bla_NDM-1_

Analysis of the WGS data, focused on the genetic environment of the *bla*_NDM-1_ of our CRAB, showed two different structures (Figure 2).

The first structure was found only in one strain which belonged to ST160 and was identical to that previously described by Poirel et al. [30]. The *bla*_NDM-1_ gene is embedded in transposon Tn*125*, a 10,099-bp composite transposon bracketed by two copies of the insertion sequence (IS) IS*Aba125* orientated in the same direction. In a search within the NCBI data, we found that the above structure is located either on the chromosome of a microorganism (AP014649, LN868200, LN997846, CP053215, CP027528, CP091596, CP130627, CP130628, CP050388, CP050403, CP072398, CP072295, CP072300, CP072290, CP072275, CP072285, CP072280, CP072270, CP072122, AND CP087312) or on plasmids (JN377410.2, CP0488828.1, CP059301, CP047975, CP098796, and CP129246).

The second structure was identified in strains which belonged to ST2, ST85, and ST2493, and its presence was previously reported, according to the NCBI data, on the chromosome of strains with origin in Saudi Arabia, Lebanon, China, India, Belgium, Spain, and the USA (CP121598, CP121595, CP121563, CP082952, CP088894, CP088895, CP121604, CP038644, CP091361, CP065392, CP060011, and CP060013). The *bla*_NDM-1_ gene in this structure is embedded in transposon 7382; Tn*7382* encompasses seven open reading frames, *cutA-tat-iso-ble_MBL_-bla*_NDM-1_-IS*Aba125*-*aph(3′)-VI*, bracketed by two direct copies of IS*Aba14* [31].

## 4. Discussion

*A. baumannii* NDM-1 producers were first detected in 2010 in India [11]. Thereafter, many reports from various parts of the world have highlighted the appearance and the dissemination of such strains internationally [11,12,13,14,15,16,17,18,19,20,21,22,23,24,25,26,32]. *A. baumannii* exhibits remarkable genetic plasticity, allowing the organism to acquire resistance genes through horizontal gene transfer. Mobile genetic elements, such as plasmids and transposons, can facilitate the spread of carbapenemase-encoding genes within and between bacterial populations [1].

Many clones and different structures have been identified in the NDM-1 producers *A. baumannii.* In the present study, two different structures (Tn*125* and Tn*7382*), all previously described, and four STs (ST160, 2, 85, and 2493), three previously reported and one newly identified, were observed in nine strains of *A.* baumannii isolated in Greece. The first structure, associated with Tn*125* (found on a ST160 strain), was found on the chromosome of several STs that originated from Vietnam (ST622), Germany (ST25, ST103, ST126, and ST267), Norway (ST374), Switzerland (ST1), Slovenia (ST25), the USA (ST1), India (ST1, ST2, and ST622), Australia (ST1), and Israel (ST2 and ST107) or on plasmids found in strains that originated from China, Malaysia, India, or France, after analysis of the NCBI data. The second structure, associated with Tn7382 (found on three ST2, three ST85, and two ST2493 strains), was detected on the chromosome of several clones that originated from Saudi Arabia (ST2, ST570), Portugal (ST85), China (ST164), Belgium (ST85), Spain (ST85), or the USA (ST570). Thus, our findings reinforce the fact that the spread of the bla*_NDM-1_* gene in *A. baumannii* is not linked to a clonal spread but to the spread of a genetic structure. A critical step in the dissemination process of the *bla*_NDM-1_ gene is the mobility of the transposon in *Acinetobacter* spp. [33].

Difficulties in treating infections caused by carbapenem-resistant *A. baumannii* stem from a multi- or extra-resistant phenotype that leaves only a handful of available antibiotics, all of which have uncertain or limited efficacy against the organism, e.g., colistin and tigecycline [34]. Cefiderocol may potentially present a therapeutic option for such infections, but the relevant clinical data are limited [35]. On the other hand, the capacity of these bacteria to survive for a long period on surfaces in the hospital environment and on medical equipment and devices can enhance their transmission during contact with contaminated surfaces or equipment, as well as though person to person, often via contaminated hands. This work re-emphasizes the need to apply diligent surveillance protocols in order to limit horizontal transfer of *bla*_NDM-1_ gene to other *A. baumannii* clones or to other recipient strains.

In conclusion, the emergence of four different clones of *bla*_NDM-1_-positive *A. baumannii* in Central Greece verifies that the dissemination of these strains was neither due to a single clone but rather to different clones carrying either the transposon Tn*125* or Tn*7382*. As part of this work, a new sequence type of the organism (ST2493) was identified. All the patients, except one who died, were discharged from the hospital after a successful treatment based mainly on the combination of colistin and tigecycline. Our surveillance program for *A. baumannii* is in progress.

## Figures and Tables

**Figure 1 microorganisms-11-02588-f001:**
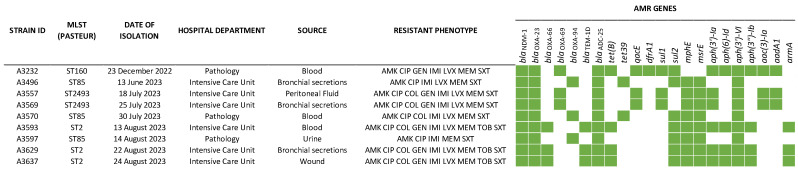
Summary presentation of antimicrobial resistance genes detected in nine *bla*_NDM-1_-positive *A. baumannii* strains (AMK: amikacin, CIP: ciprofloxacin, COL: colistin, GEN: gentamicin IMI: imipenem, LVX: levofloxacin, MEM: meropenem, SXT: trimethoprim–sulphamethoxazole).

**Figure 2 microorganisms-11-02588-f002:**
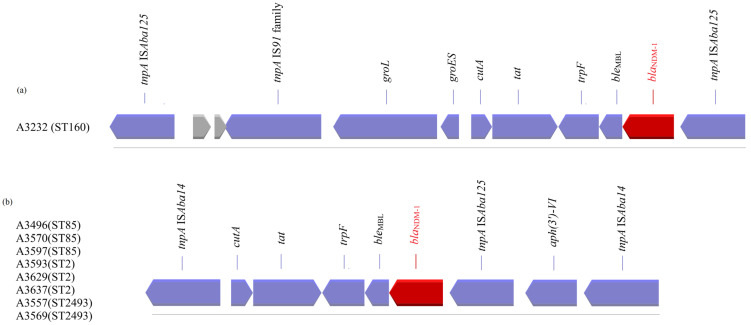
Summary presentation of transposons (**a**) Tn*125* and (**b**) Tn*7382*, carrying the *bla*_NDM-1_ gene detected in nine *A. baumannii* strains (coding sequences (blue/red colour) are labelled by their gene, and sequences encoding hypothetical proteins are presented in grey).

## Data Availability

The genomes of the *bla*_NDM-1_-positive *A. baumannii* isolates (A3232, A3496, A3557, and A3569) have been deposited in GenBank under BioProject accession PRJNA1018864.

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
