# Peer review of "The Molecular Characterization of blaNDM-1-Positive Acinetobacter baumannii Isolated in Central Greece"

_microorganisms, 2023, doi:10.3390/microorganisms11102588_

Round 1

Reviewer 1 Report

in the materials and methods 

 why Tigycycline senstivity (MIC  ) was not mentioned 

Author Response

Reviewer 1: “in the materials and methods why Tigycycline senstivity (MIC  ) was not mentioned

Response: Thank you for your comment. The MIC values of tigecycline in these 9 isolated range from 1 to 4 mg/L. However, in the present context and in line with EUCAST guidelines, there is currently insufficient evidence to support the effectiveness of tigecycline against A. baumannii strains. Therefore, it is not possible to draw any conclusions or interpretations regarding tigecycline values. 

Reviewer 2 Report

The manuscript of Tsilipounidaki et al. “Molecular characterization ...”  needs small changes:

1.      Line 68 MIC colistin – please add company, whether the Authors use commercial test or no – this information should be add

2.      Line 69 EUCAST guideline should be cited in references

3.      In the results section there is susceptibility to amikacin, ciprofloxacin, trimetoprim-sulphametoxazole, genatmicin, but there is no information about the method in section Materials and Methods

4.      Lines 100- 105 the unit is mg L-1, should be mg/L

Author Response

Reviewer 2: “Line 68 MIC colistin – please add company, whether the Authors use commercial test or no – this information should be add”

Response: Thank you for your comment. The information has been added (line 73).

“Line 69 EUCAST guideline should be cited in references”

Response: Thank you for your comment. The citation has been added (Reference number 27).

” In the results section there is susceptibility to amikacin, ciprofloxacin, trimetoprim-sulphametoxazole, genatmicin, but there is no information about the method in section Materials and Methods”

Response: The susceptibility testing to these antimicrobials was performed by automated system Vitek II (line 69-70).

“Lines 100- 105 the unit is mg L-1, should be mg/L”

Response: Thank you for your comment. The unit has been changed.

Reviewer 3 Report

In this manuscript, the authors characterize by WGS four clinical carbapenem-resistant NDM positive A. baumannii isolates.The manuscript is well written, concise and results emphasise A. baumannii genetic plasticity mediated by transposons in this case.

minor comments

The authors could refer to the number of A baumannii clinical isolates assessed and the percentage of blaNDM carriage .

Author Response

Reviewer 3: “The authors could refer to the number of A baumannii clinical isolates assessed and the percentage of blaNDM carriage.”

Response: Thank you for your comment. The total number of isolates examined during the study period was 171, and among them 9 have been found to carry blaNDM-1.

Reviewer 4 Report

Great job on your clear and straightforward presentation and writing of the manuscript.

I would advise providing feedback if the scenario has changed and the patients have benefited from the antibiotics being used, if issues have arisen, or if multiple bacteria have been found at the same time.

Minor editing of English language are required

Author Response

Reviewer 4: “I would advise providing feedback if the scenario has changed and the patients have benefited from the antibiotics being used, if issues have arisen, or if multiple bacteria have been found at the same time.”

Response : Thank you for your comment. All the patients , except one who died, were discharged from the hospital after a successful treatment based on combination of colistin plus tigecycline. Our surveillance program for A. baumannii is in progress. A sentence was added in the Conclusion.